# Exploitation of Quercetin’s Antioxidative Properties in Potential Alternative Therapeutic Options for Neurodegenerative Diseases

**DOI:** 10.3390/antiox12071418

**Published:** 2023-07-13

**Authors:** Viorica Rarinca, Mircea Nicusor Nicoara, Dorel Ureche, Alin Ciobica

**Affiliations:** 1Doctoral School of Geosciences, Faculty of Geography and Geology, Alexandru Ioan Cuza University of Iasi, No 20A, Carol I Avenue, 700505 Iasi, Romania; rarinca_viorica@yahoo.com; 2Doctoral School of Biology, Faculty of Biology, Alexandru Ioan Cuza University of Iasi, No 20A, Carol I Avenue, 700506 Iasi, Romania; 3Department of Biology, Faculty of Biology, Alexandru Ioan Cuza University of Iasi, No 20A, Carol I Avenue, 700505 Iasi, Romania; 4Department of Biology, Ecology and Environmental Protection, Faculty of Sciences, University Vasile Alecsandri of Bacau, Calea Marasesti Street, No 157, 600115 Bacau, Romania; 5Center of Biomedical Research, Romanian Academy, No 8, Carol I Avenue, 700506 Iasi, Romania; 6Academy of Romanian Scientists, No 54, Independence Street, Sector 5, 050094 Bucharest, Romania

**Keywords:** oxidative stress, neurodegenerative diseases, quercetin, neuroprotective effects

## Abstract

Oxidative stress (OS) is a condition in which there is an excess of reactive oxygen species (ROS) in the body, which can lead to cell and tissue damage. This occurs when there is an overproduction of ROS or when the body’s antioxidant defense systems are overwhelmed. Quercetin (Que) is part of a group of compounds called flavonoids. It is found in high concentrations in vegetables, fruits, and other foods. Over the past decade, a growing number of studies have highlighted the therapeutic potential of flavonoids to modulate neuronal function and prevent age-related neurodegeneration. Therefore, Que has been shown to have antioxidant, anticancer, and anti-inflammatory properties, both in vitro and in vivo. Due to its antioxidant character, Que alleviates oxidative stress, thus improving cognitive function, reducing the risk of neurodegenerative diseases. On the other hand, Que can also help support the body’s natural antioxidant defense systems, thus being a potentially practical supplement for managing OS. This review focuses on experimental studies supporting the neuroprotective effects of Que in Alzheimer’s disease (AD), Parkinson’s disease (PD), Huntington’s disease (HD), and epilepsy.

## 1. Introduction

Excessive OS is the result of disturbing the balance between oxidation and antioxidant systems, with a tendency to oxidize. OS can cause many reactions, such as protease stimulation, neutrophil infiltration, and the explosion of oxidative intermediates [1]. In addition, OS is thought to play a key role in the progressive degeneration and/or death of nerve cells, especially in neurodegenerative diseases, where it acts as a mediator of the side effects of several neurotoxic substances and as a mechanism of age-related degenerative processes [2]. The ROS scavenger is often used to counteract the effects of OS in neurons [3,4]. Numerous studies have shown that Que, by eliminating oxygen radicals and via metal-chelating operations, attenuates the neuronal damage mediated by OS [5,6].

To survive, aerobic multicellular organisms need molecular oxygen (O_2_), to the detriment of oxygen, which is susceptible to radical formation due to its electronic structure. Reactive oxygen species are the natural by-products of normal oxygen metabolism and play significant roles in homeostasis and cellular signaling. OS increases in the cellular environment when oxygen homeostasis is not maintained. ROS are oxygen free radicals or small molecules derived from oxygen, such as peroxyl radical (ROO•), hydroxyl radical (•OH), superoxide anion (•O_2_^−^), and alkoxyl (RO^−^). ROS could also come from non-radicals such as ozone (O_3_), hypochlorous acid (HOCl), singlet oxygen (^1^O_2_), and hydrogen peroxide (H_2_O_2_). These non-radicals are oxidizing agents or are easily converted to radicals [7,8].

Neurological diseases are a consequence of genetics, environmental factors, and even age [9]. During periods of environmental stress, such as ultraviolet A (UVA) and ultraviolet B (UVB) radiation, exposure to heat, and ionizing radiation, their levels could increase dramatically. A study by Erden et al. in 2001 [10] showed that exposure to UVA radiation can induce ROS production, leading to damage to cellular elements, but can also induce the benefits that antioxidant Que brings to cells, protecting them from the harmful effects of radiation.

Neurodegeneration is characterized by progressive deterioration of the structure and function of neurons and is accompanied by severe cognitive deficits. Aging is the main risk factor for neurodegenerative disorders such as AD, PD, and HD. Mitochondrial dysfunction and OS also trigger neurodegeneration. Recent studies have supported the mechanisms by which Que supports brain health [11].

The first studies involving flavonoids to reduce OS were performed at the end of the twentieth century [12]. Numerous in vitro and in vivo studies have reported the neuroprotective properties of Que [11,13,14]. Thus, it has been observed to protect neurons from oxidative damage and reduce lipid peroxidation as well (see Figure 1). On the other hand, in addition to its antioxidant properties, Que can inhibit the formation of amyloid-β proteins in the fibrils, counteracting cell lysis and inflammatory cascade [13]. Flavonoids, but also foods containing flavonoids, can have multiple beneficial effects on the treatment of conditions involving OS, such as AD, PD, aging itself, atherosclerosis, and ischemia [12,15,16].

Que, or 2-(3,4-dihydroxy phenyl)-3,5,7- trihydroxychromen-4-one, is commonly found in our diet and is found in abundance as a secondary metabolite in vegetables and fruits. According to the USDA (United States Department of Agriculture) database on flavonoid concentration in foods [18] and Table 1, the highest Que concentrations are in capers, dill weed, oregano, onions, cranberries, cherries, and red fruits; in addition to fruits and vegetables, they are also found in beverages such as red wine and black tea. It has also been isolated and marketed as a dietary supplement, in the form of free aglycone, used in doses of 1000 mg per day, exceeding the usual levels of food intake, which is 200–500 mg per day in individuals who consume large amounts of vegetables and fruits [18,19].

Que is a more potent antioxidant than other antioxidant nutrients, such as vitamin C, vitamin E, and β-carotene [33]. Due to the five hydroxyl groups present in its structure that can bind to ROS, Que has a higher antioxidant potential than many other flavonoids [34,35]. In addition to its antioxidant activity, Que has anti-cancer effects [36,37,38,39,40]; anti-inflammatory [34,41,42,43], antiviral [44,45,46], and antibacterial properties [47,48,49,50,51]; cardioprotective effects [52]; and neuroprotective effects vs. brain ischemia [53,54].

In many studies, Que is also reported to have adverse effects, such as the induction of mutations, chromosomal aberrations, and single-stranded deoxyribonucleic acid (DNA) ruptures in various eukaryotic cell systems in vitro [55].

Que is less toxic than curcumin or gallic acid due to the LD_50_ value of 484 µg mL^−1^, while the LD_50_ value for curcumin is 135 µg mL^−1^ and that for gallic acid is 304 µg mL^−1^ [56].

This review focuses on the preventive and therapeutic capacity of Que in neurological and neurodegenerative diseases along with its potential mechanisms of action. Furthermore, we also summarized the biological sources and other pharmacological activities of this antioxidant compound.

Que acts as a protector of neurons against severe OS, but also against free radical attack by easily intercalating its molecules in DNA, thus forming a protective barrier against stronger intercalators and/or ROS attack [57].

## 2. Methodology


**Search Strategy**


The current systematic review was conducted following the Preferred Reporting Items for Systematic Reviews and Meta-Analysis (PRISMA) guidelines [58], employing several electronic databases (Science Direct, PubMed, and Google Scholar) and using the following keywords: ((quercetin[Title/Abstract]) AND (Alzheimer’s disease[Title/Abstract])) AND (amyloid[Title/Abstract]); (quercetin[Title/Abstract]) AND (Parkinson’s disease[Title/Abstract]); (quercetin[Title/Abstract]) AND (Huntington’s disease[Title/Abstract]); (quercetin[Title/Abstract]) AND (pentylenetetrazol [Title/Abstract]). Inclusion criteria included experimental studies (in vivo and in vitro) until May 2023 in English, which evaluated the antioxidant properties of quercetin in potential alternative therapeutic options for neurodegenerative diseases.


**Exclusion Criteria**


We applied the following exclusion criteria: (1) conference abstracts, books, book chapters, and unpublished results; (2) non-English papers; (3) reviews, systematic reviews, meta-analyses, and letters.


**Data Extraction**


Among the initial 868 reports that were collected through electronic search, 554 were omitted due to duplicated results, 47 were ruled out because of the article type, and 190 review articles were omitted and were deemed irrelevant based on abstract and/or title information. Additionally, 1 was excluded because it was not in the English language.


**Data Synthesis**


Finally, 43 articles were included in this study, as demonstrated in a diagram of the literature search and selection process (see Figure 2). It was thought that the studies would be too heterogeneous to be combined. Therefore, a narrative synthesis was performed. The results are summarized according to the type of neurodegenerative disease (AD, PD, HD, and epilepsy) and the test (in vivo, in vitro).

## 3. Oxidative Stress

OS is a condition in which there is an imbalance between the production of ROS and the body’s ability to neutralize them, causing damage to lipids, proteins, and DNA. ROS are naturally produced by the body as a by-product of normal metabolism but can also be produced by environmental factors such as exposure to pollution, radiation, and certain chemicals. In the context of neurological disease, OS has been shown to play a role in the development and progression of several conditions.

Recent studies have shown that Que can protect against OS-induced cell death by inhibiting the activity of caspase-3 [59] and increasing the body’s antioxidant capacity by regulating glutathione (GSH) levels [60]. Que has also been shown to reduce inflammation and OS and improve wound healing in animal models of Alzheimer’s disease [43] and reduce inflammatory pain by inhibiting OS pathways [61] (see Figure 2). These findings suggest that Que may be beneficial in reducing OS and inflammation in humans [62].

In AD, OS has been shown to accelerate the formation of amyloid plaques in the brain, which are a hallmark of the disease. Also, in PD, OS can cause the death of dopaminergic neurons, leading to loss of motor function. In HD, stress can increase levels of the glutamate neurotransmitter in the brain, which can lead to neuron death and worsening symptoms. Thus, in Figure 3, we highlight the enzymes that have the role of protecting the body from OS and have neuroprotective effects in the case of neurodegenerative diseases.

### 3.1. Alzheimer’s Disease

Alzheimer’s disease is the most common neurodegenerative disease, accounting for about two-thirds (60–80%) of all cases of dementia, and it affects mainly the elderly (aged 65 or older) [63]. The pathogenesis of AD is commonly associated with the extracellular accumulation of amyloid-β (1–40, 1–42) aggregates and the hyperphosphorylation of tau proteins, leading to neurofibrillary tangles (NFT) and synaptic dysfunction [13,64,65,66]. An estimated 44 million people worldwide are affected by AD or a related form of dementia, with a prevalence rate of 4.6 million new cases each year. The prevalence rate of AD increases with age: the rate doubles every 5 years from the age of 60 [67,68].

OS plays an important role in AD, which, through ROS generation, can amplify or initiate the disease (Table 2). The reduction reaction of hydrogen peroxide results in the production of reactive oxygen species, thus damaging brain tissue and disrupting brain-cell repair [69,70,71]. Thus, the administration (ad) of Que before the treatment decreases the damage to the cell membrane induced by the OS caused by H_2_O_2_ [70].

AD is characterized by neuronal loss, which is preceded by the extracellular accumulation of Aβ_1–40_, and Aβ_1–42_. Antioxidants such as Que increase the resistance of neurons to OS by modulating cell-death mechanisms. Thus, Que protects the mouse hippocampal cell line HT-22 from glutamate-induced oxidative toxicity and lipid peroxidation by blocking the production of free radicals [69]. Also, pretreatment of primary hippocampal cultures with Que significantly attenuated Aβ_1–42_-induced cytotoxicity, protein oxidation (protein carbonyl, 3-nitrotyrosine), lipid peroxidation (protein-bound 4-hydroxy-2-nonenal), and apoptosis. There were also observed protective effects against Aβ_1–42_ toxicity by modulating OS at lower concentrations (5 and 10 μM), while in the cases of higher concentrations (20 and 40 μM), the effects were not only non-neuroprotective, but toxic [13].

However, recent research shows that the more hydroxyl groups the structure of the molecule contains, the stronger the anti-amyloidogenic activity. Therefore, one of the potential benefits of Que is its ability to act as an anti-amyloidogenic agent due to its five hydroxyl groups (see Figure 1), which means it can prevent the formation of amyloid plaques in the brain [72,73].

Numerous studies demonstrate anticancer and apoptosis-inducing effects in vitro on a variety of cancer cell lines, including murine neuroblastoma HT-22 cells or human-brain microvascular endothelial cells [69,74,75,76]. Que was found to be non-cytotoxic and strongly protected HT-22 cells from fibril formation [76]. In addition, Ishige et al. in 2001 [69], using the HT-22 mouse hippocampal cell line, found three distinct mechanisms of flavonoid protection in cell death, including increased intracellular GSH, a direct decrease in ROS levels, and the prevention of Ca^2+^ influx.

In a study on the stable cell line of the Swedish mutant of amyloid precursor protein (APP695-transfected SH-SY5Y), no effects of Que were observed in the middle-late stage of AD; instead, effects were observed in the mid-early stage, when the reduction in β-amyloid-converting enzyme 1 (BACE1) activity was recorded [73].

The optimal concentration of Que required for the effective destabilization of Aβ fibrils has been found to be in the range of 0.1–1 μM [74,77]. Thus, in a study using neurons of the rat hippocampal region, optimal doses of Que administration were beneficial for protecting against Aβ_25–35_-induced amnesic injury by reducing lipid peroxidase, ROS, and GPx [77]. In another study on pheochromocytoma (PC12) cells, Que was found to increase the survival rate of H_2_O_2_-damaged cells, decrease lipid peroxidation and GSH level, and provide mitochondrial protection mechanisms [70]. On the other hand, Yu et al. in 2020 [78] showed that Que has beneficial effects, so it can increase the PC12 cell-survival damaged by Aβ_25–35_, antagonize the toxicity of Aβ, promote cell proliferation, and provide some neuroprotective effects.

Also, in a study involving homozygotic transgenic mouse line B6.129S7-Sod2tm1Leb/J, where Que was administered orally at a dose of 50 mg kg^−1^ body weight (b.w.) twice a week for four weeks, the results showed that Que had a protective effect against hydrogen peroxide- and paraquat-induced OS in the mice [71]. In addition, Que has the effect of reducing anion superoxide levels that increased with H_2_O_2_ and Aβ treatment in hippocampal neurons or human-brain microvascular endothelial cells (hBMECs) [71,75].

Quercetin-3-glucuronide (Q3G), a glucuronide conjugate of Que, has been identified as a potential intervention for AD due to its ability to target the brain. Thus, several studies have shown that Q3G may be able to alleviate neuroinflammation and reduce OS in nerve cells, both of which are associated with AD [79,80]. Additionally, Q3G has been found to specifically localize in human brain tissue, suggesting that it may be able to cross the blood–brain barrier (BBB) and reach areas of the brain affected by AD [79]. The mechanism by which Que can cross the BBB is through passive diffusion; this is a process that allows small molecules to move across the BBB from an area of high concentration to an area of low concentration [81]. Thus, Ho et al. in 2013 [82] observed that Q3G significantly reduced the generation of β-amyloid peptides using cultures of primary neurons generated by the mouse model Tg2576 AD.

A high concentration of Que was found in *Ginkgo biloba*, thus showing that *Ginkgo biloba* extract (EGb761) and its constituents, Que and ginkgolide B, have protective effects against the cytotoxic action of Aβ_1–42_ via intraperitoneal administration (i.p.), thereby ameliorating the oxidative phosphorylation deficits, and mitochondrial dysfunction in AD [83]. Another plant used in pharmacotherapy and in which Que is found is *Acanthopanax henryi*, which can potentiate cholinergic activity by inhibiting acetylcholinesterase (AchE) [84].

**Table 2 antioxidants-12-01418-t002:** Protective effects against OS, neuroinflammation, and Aβ accumulation induced by Que in vitro.

Types of Que	Concentration of Que	Model	Exposure	Effects	Ref.
Que	Dosage: 2.2 μM; Duration: 24 h;	HT-22 mouse hippocampal cell	H_2_O_2_	↓ lipid peroxidation, ↑ intracellular GSH, ↓ ROS	[69]
	Dosage: 10–100 µmol L^−1^; Duration: 10 min;	PC12 cells	H_2_O_2_	↓ lipid peroxidation, ↓ GSH, mitochondrial protection mechanisms	[70]
	Dosage: 50 mg; kg^−1^ b.w.; Duration: 2 times a week for 4 weeks;	homozygotic transgenic mouse line B6.129S7-Sod2tm1Leb/J	H_2_O_2_ and Aβ	↓ ROS levels, improved the typical morphology of mitochondria, prevented mitochondrial dysfunction	[71]
	Dosage: 10 μM;	APP695-transfected SH-SY5Y cells	Aβ_25–35_	↓ ROS, ↓ BACE, ↓ Aβ, ↓ GSH, ↓ lipid peroxidation	[73]
	Dosage: 10 and 50 μM; Duration: 7 days;	hek cells	Aβ_1–42_ or Aβ_1–40_	↓ Aβ peptides, ↓ the performed mature fibrils	[74]
	Dosage: 5 or 10 mg kg^−1^ b.w.; Ad: p.o.; Duration: once daily;	hBMECs	fAβ_1–40_	↓ SOD, ↓ LDH	[75]
	Dosage: 2.4 µg mL^−1^;	HT-22 murine neuroblastoma cells	Aβ_25–35_	↓ amyloidogenic Aβ peptides, inhibited Aβ fibril formation.	[76]
	Dosage: 10, 20, 40, and 80 μmol L^−1^; Duration: 24 h, 48 h, and 72 h;	PC12 cells	Aβ_25–35_	↑ the survival rate of PC12 injured by Aβ_25-35_, promoted cell proliferation, and antagonized the toxicity of Aβ, ↓ ROS	[78]
Q3G	Dosage: 25 μmol L^−1^;	Tg2576 AD primary neuron cultures	Aβ_1–40,_ Aβ_1–42_	↑ neuronal survival, ↑ c-Jun N-terminal kinases, ↓ stress-induced impairments	[82]
Que/*Ginkgo biloba*	Dosage: 1.5–6 μg mL^−1^;	SHSY5Y human neuroblastoma cells	Aβ_1–42_	↓ Akt signaling pathways, ↓ Aβ toxicity, ↓ platelet-activating factor	[83]
Que/ *Acanthopanax henryi*	Dosage: 2.5, 5, 10, 20, and 40 μg mL^−1^;	cell-free system		↓ AchE activity, ↑ antioxidant activity	[84]

Abbreviations: ↑, increase; ↓, decrease; Aβ, amyloid beta-peptide; AchE, acetylcholinesterase; BACE, β-amyloid-converting enzyme 1; b.w., body weight; CAT, catalase; GPx, superoxide dismutase; GSH, glutathione; hBMECs, human-brain microvascular endothelial cells; hek, human embryonic kidney; LDH, lactate dehydrogenase; OS, oxidative stress; PC12, pheochromocytoma; Q3G, quercetin-3-glucuronide; Que, quercetin; ROS, reactive oxygen species; SOD, superoxide dismutase.

In mitochondria, the first free radical to form is the superoxide radical, which is catalyzed by superoxide dismutase (SOD) and can cause irreversible damage to nucleic acids, proteins, phospholipids, and/or signaling pathways, thus contributing to apoptosis and intoxication [85].

In vivo studies (Table 3) in triple transgenic mice models of AD (3xTg-AD) have shown that Que can disaggregate amyloid fibrils, such as extracellular amyloid β-peptide, tauopathy astrogliosis, and microgliosis in the hippocampus and amygdala, and improves their spatial memory and learning [86,87]. Additionally, the results showed that Que tended to improve active behaviors of 3xTG-AD mice and decreased neurodegeneration markers in mice [87]. Additionally, in the case of APPswe/PS1dE9 transgenic mice, it was observed that long-term Que consumption prevents memory loss, Aβ-induced neurotoxicity, and mitochondrial dysfunctions [88].

Furthermore, Hayakawa et al. in 2015 [89] showed that Que has memory-enhancing effects in older mice and delays the deterioration of memory in the early stages of Alzheimer’s, since it reduces eIF2a and ATF4 expression by inducing GADD34 in the brain. Also, Que can partially block the effect of other genes that play an important role in Alzheimer’s disease, such as tumor necrosis factor-alfa (TNF-α), interleukin 1 beta (IL-1β), and interleukin 6 (IL-6) [90].

On the other hand, oral (p.o.) treatment with 500 mg kg^−1^ b.w. Que for 10 days can significantly increase brain apoE levels and reduce insoluble Aβ levels in the cortex of 5xFAD amyloid-model mice [91].

Interestingly, this memory impairment was markedly ameliorated by oral treatment with Que nanoencapsulated in zein nanoparticles (25 mg kg^−1^ every 48 h for 2 months), while the administration of free Que was not able to reverse the faulty behavior, despite a higher administration frequency [92]. Also, pretreatment with Que decreases the effects induced by Aβ_1–42_ in adult male Sprague Dawley rats [93].

Scopolamine administration causes short-term and long-term memory loss because it blocks muscarinic cholinergic receptors in the brain and interferes with learning and memory [94,95]. There are studies that have found that Que alleviates scopolamine-induced memory deficits by protecting against neuroinflammation and neurodegeneration by inhibiting OS and acetylcholinesterase activity, reverses synaptic loss in the cortex and hippocampus of the brain of adult mice, and suppresses memory impairment [94,95].

Aluminum is a toxic metal that has neurological effects, including Alzheimer’s disease, by generating ROS [96]. Increased production of reactive oxygen species leads to the disruption of cellular antioxidant defense systems and to the release of cytochrome c from the mitochondria into the cytosol, resulting in apoptotic cell death [96,97]. Thus, the administration of 10 mg kg^−1^ b.w Que reduces the effects induced by aluminum, thus reducing OS. In addition, it prevents cytochrome c translocation [96].

In addition, Hou et al. in 2010 [98] showed that flavonols can antagonize the toxicity of Aβ and improve the expression of brain-derived neurotrophic factor (BDNF) in the hippocampus of double transgenic mice.

**Table 3 antioxidants-12-01418-t003:** Protective effects against oxidative stress, neuroinflammation, and Aβ accumulation induced by Que in vivo.

Types of Que	Concentration	Model	Exposure	Effects	Ref.
Que	Dosage: 25 mg kg^−1^ b.w.; Ad: i.p.; Duration: every 2 days for 3 months;	3xTg-AD mice		↓ tauopathy, ↓ β-amyloidosis, ↑ memory, ↑ learning, ↓ microgliosis, ↓ astrogliosis	[86]
	Dosage: 100 mg kg^−1^ b.w.; Ad: gavage; Duration: every 48 h for 12 months;	3xTg-AD mice		↓ neurodegeneration, ↓ β-amyloidosis	[87]
	Dosage: 20 and 40 mg kg^−1^ b.w.; Ad: p.o.; Duration: 16 weeks;	adult male C57BL mice		↑ MMP, ↑ ATP levels, ↓ ROS	[88]
	Dosage: 20 mg; Ad: p.o.; Duration: 5 weeks;	APP23 AD mice model	Aβ	↓ eIF2α, ↓ ATF4, ↓ GADD34, ↑ memory in aged mice, ↓ memory deterioration in the early stage of AD, ↓ memory dysfunction, ↓ OS	[89]
	Dosage: 1% in mouse chow; Ad: p.o.; Duration: from 3 to 13 months;	double transgenic female mice		↓ neuroinflammation, ↓ neurodegeneration, ↓ IL-1β	[90]
	Dosage: 25 mg kg^−1^; Ad: p.o.; Duration: 2 times a week for 2 months;	SAMP8 mice		↑ the cognition and memory impairments, ↓ astrogliosis	[92]
	Dosage: 100 mg kg^−1^ b.w.; Ad: p.o.; Duration: 22 days;	adult male Sprague Dawley rats	Aβ_1–42_	↑ expression of Nrf2/HO-1 in rat brains, ↓ Aβ_1-42_ level, ↓ antioxidant activity	[93]
	Dosage: 12.5 and 25 mg kg^−1^;	mice	Scopolamine	↓ OS, ↓ AchE activity	[94]
	Dosage: 30 mg kg^−1^ b.w.; Ad: i.p.; Duration: every day for 8 days;	male albino Wistar rats	Scopolamine	abridged transfer latency, ↓ avoidance response, ↓ 3,4-methylenedioxyamphetamine, acetylcholinesterase levels, ↑ CAT, ↑ GSH levels	[95]
	Dosage: 10 mg kg ^−1^ b.w.; Ad: p.o.; Duration: every day for 12 weeks;	male albino Wistar rats	aluminum	↓ ROS production, ↑ mitochondrial superoxide dismutase activity	[96]
Que/ginkgo flavonols	Dosage: 4.8% in extract, all based on weight;	double transgenic (TgAPP/PS1) mice	-	reversed the spatial learning deficit	[98]

Abbreviations: ↑, increase; ↓, decrease; Aβ, amyloid beta-peptide; AchE, acetylcholinesterase; ATP, adenosine triphosphate; b.w., body weight; CAT, catalase; GPx, glutathione peroxidase; GSH, glutathione; hBMECs, human-brain microvascular endothelial cells; hek, human embryonic kidney; IL-1β, interleukin 1 beta; i.p., intraperitoneal; LDH, lactate dehydrogenase; MMP, matrix metalloproteinases; OS, oxidative stress; PC12, pheochromocytoma; Q3G, quercetin-3-glucuronide; Que, quercetin; ROS, reactive oxygen species; SOD, superoxide dismutase.

### 3.2. Parkinson’s Disease

Parkinson’s disease is the second most common neurodegenerative disorder worldwide, affecting 1% of the global population aged 65 years and older; it has significant morbidity and mortality [99]. An increasing percentage of research indicates the association of PD with microglial activation, resulting in an increase in various inflammatory mediators and neuroinflammation [100,101]. 1-Methyl-4-phenylpyridinium (MPP+) is the ultimate toxic agent formed by the metabolism of MPTP and can activate glial cells to induce neuroinflammation [102]. Research has shown that MPP+ induces microglial activation and the degeneration of dopaminergic neurons (Table 4), as well as the generation of ROS in dopaminergic neurons [103]. On the other hand, Que administration protects microglia cells against MPP+-induced increases in the mRNA and protein levels of IL-1, IL-6, and TNF-α, due to its antioxidant action [102]. In addition to the loss of dopaminergic neurons in the substantia nigra pars compacta, PD is also characterized by the abnormal accumulation and aggregation of α-synuclein (α-Syn) in the form of Lewy bodies [104,105]. Thus, the formation of α-Syn fibrils can be inhibited by Que and oxidized by Que through their 1:1 covalent binding [105].

Another neurotoxic synthetic organic compound used by researchers to selectively destroy dopaminergic and noradrenergic neurons is 6-hydroxydopamine (6-OHDA) [106,107]. It is a hydroxylated analogue of dopamine and is a benzenetriol with hydroxyl groups on the phenyl ring at positions 2, 4, and 5. Isoquercetin, a flavonol derived from Que, has also been found to have protective effects against 6-OHDA-induced oxidative damage in a rat model of PD. They observed that antioxidant enzymes, catalase (CAT), SOD, GPX, and GSH levels, which were previously attenuated by 6-OHDA, increased significantly [106,107].

In vitro studies have shown that Que can improve mitochondrial quality control, reduce OS, and increase levels of antioxidant enzymes (Table 5) [102,105,106,107]. Instead, in vivo studies in mice and 6-OHDA-induced PD rat models demonstrated that Que can improve locomotor and muscle activity, increase striatal dopamine levels, and protect neurons from mitochondrial dysfunction [108,109,110,111,112].

Studies have shown that Que has neuroprotective effects against MPTP-induced neurotoxicity in Wistar rats and adult male C57BL/6 mice [108,109,110]. Que was found to reduce OS and neuroinflammatory cytokines in rats [108,109], as well as restore motor and non-motor symptoms (depression and cognitive impairment) of PD in rats injected with rotenone [113,114]. Additionally, Que supplementation was found to improve striatal cholinergic function and reduce rotenone-induced OS in rats [114].

On the other hand, the administration of fish oil can attenuate rotenone-induced oxidative impairments and mitochondrial dysfunctions in rat brains [115]. Combined oral supplementation with fish oil and Que has been found to enhance neuroprotection in a chronic rotenone rat model, suggesting potential relevance for PD [115].

### 3.3. Huntington’s Disease

Along with AD and PD, HD is a major health problem worldwide, with a major financial impact [116]. HD is an autosomal dominant inherited disorder, the treatment of which is clinically available but provides only symptomatic relief. These drugs are available by prescription and have side effects such as anxiety and depression.

For an experimental model of HD, 3-nitropropionic acid (3-NPA) was administered, which altered the mitochondrial metabolism, decreased cellular ATP level, and included nerve-cell death by increasing OS.

In a study by Sandhir and Mehrotra [117] in which female Wistar rats were used as model organisms and in which Que was orally administered at a dose of 25 mg kg^−1^ for 21 days, for 17 of these 21 days concomitantly with 3-NPA, an attenuation of motor deficits was observed which were assessed using the narrow-beam walking test and fingerprint analysis. Furthermore, molecular changes induced by 3-NPA acid were observed, which were reversed, thus increasing the level of OS and lowering the ATP concentration [117].

On the other hand, a study by Chakraborty [118] failed to confirm the beneficial effect of Que on the 3-NP-induced striatal neuronal lesion (Table 6). However, the conditions of the two studies varied little. Chakraborty used male rats as model organisms, the duration of administration of 3-NP and Que was 4 days, and the concentration was higher (25–50 mg kg^−1^) than that which was administered by Sandhir and Mehrotra [117], where the dose was administered subchronically (25 mg kg^−1^). Although Que had no effects on 3-NP-induced striatal neuronal injury, it significantly attenuated neurotoxin-induced anxiety, decreased microglial proliferation, and increased the number of astrocytes in the lesion core [118].

In addition, Que in combination with other antioxidants, such as lycopene, decreases anxiety and depression [119]. Furthermore, the use of dietary antioxidants as adjuvants with n-3 fatty acids is increasingly being used, as they offer a higher degree of protection. Thus, the efficacy of Que in combination with fish oil was observed in a rat model previously treated with 3-NPA, where it decreased OS and improved motor function [120].

Quinolinic acid (QA) is also a paradigm of HD, and co-administration of antioxidants such as Que with sesamol minimizes neurochemical, behavioral, and biochemical alterations in rat brains [121]. However, these data appear inconsistent and unequivocal conclusions cannot be drawn.

### 3.4. Epilepsy

Epilepsy is a neurological disorder characterized by recurrent spontaneous seizures (Table 7), being caused by an imbalance in excitatory and inhibitory neurotransmission [122]. Glutamate and γ-amino butyric acid (GABA) are the major excitatory and inhibitory neurotransmitters in the CNS [123]. A GABA receptor antagonist is pentylenetetrazol (PTZ), which is used to create a chemically induced seizure model in animals [123]. The frequency and severity of these recurrent seizures can vary. Thus, a low dose of Que (25 mg kg ^−1^) can reduce the number and duration of spike-wave discharges in WAG/Rij rats [124]. In addition, a reduction in the levels of TNF-alpha, IL-6, and NO was observed compared with the control group.

In a study using PTZ-induced seizure model rats, Que administration at 10 mg kg^−1^ intraperitoneally 30 min before PTZ injection significantly prolonged the onset and reduced the severity of the seizure, but, at an increased concentration of 40 mg kg^−1^, Que failed to prevent the effects of PTZ [123]. Also, Nassiri-Asl et al. [125] showed that the administration of 35 mg kg^−1^ PTZ after 50 mg kg^−1^ Que reduces seizure severity during kindling and improves performance in a passive avoidance task in kindled rats. Also, supplementation of levetiracetam with quercetin improved depression that is associated with epilepsy, led to decreased immobility time and reduced seizure severity. [126]. In addition, Choudhary et al. in 2011 [127] isolated and evaluated the antiepileptic potential of both the acute and chronic flavonoid fractions of *Anisomeles malabarica* leaves. Toxic effects were observed In the acute treatment (25 and 50 mg kg^−1^, i.p.) while for the chronic treatment for one week (6.25 and 12.5 mg kg^−1^, i.p.) a significant antiepileptic effect was observed, without causing neurotoxic side effects [127].

## 4. Conclusions

Research on Que’s neuroprotective effects suggests that it can be used to protect against various neurodegenerative diseases. Que has been shown to reduce OS and inflammation, which are both associated with neurodegeneration. It has also been found to reduce hippocampal tau phosphorylation, which is a marker of AD.

In addition, Que has been found to act through numerous mechanistic targets to provide neuroprotection, including the modulation of receptor pathways. It has also been found to have protective effects when combined with vitamin C, lycopene, fish oil, or sesamol.

Overall, research suggests that Que may be an effective agent for the prevention of progressive age-related neurodegenerative diseases such as AD, PD, HD, and epilepsy, respectively. However, more research is needed to draw more concrete conclusions about the efficacy of Que in these disorders.

## Figures and Tables

**Figure 1 antioxidants-12-01418-f001:**
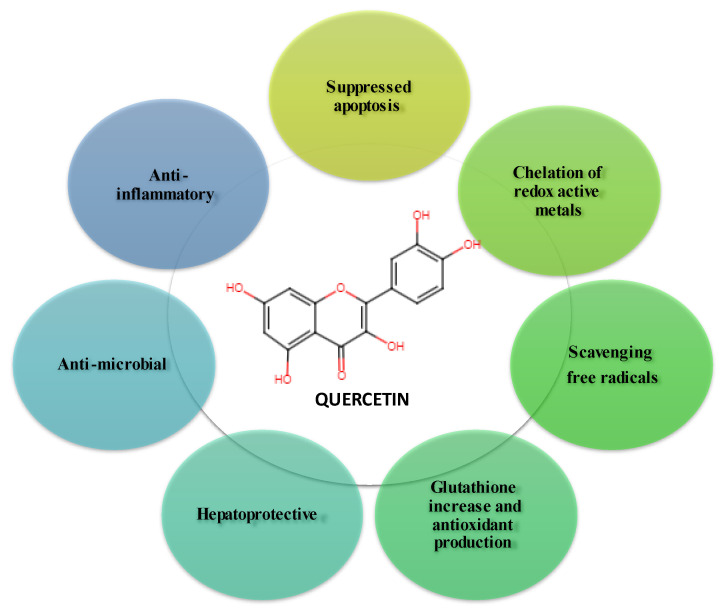
Main effects of Que [14,16,17].

**Figure 2 antioxidants-12-01418-f002:**
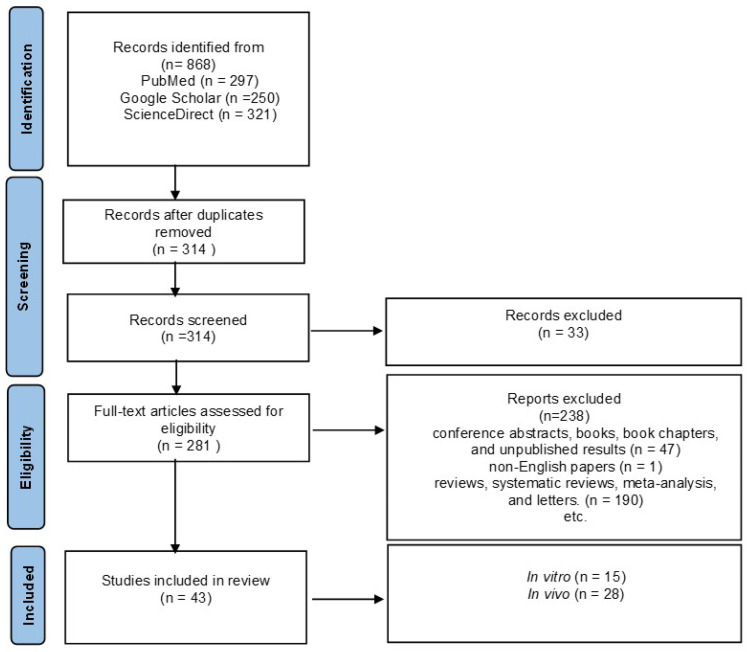
The PRISMA flow chart of the selection process for the included studies.

**Figure 3 antioxidants-12-01418-f003:**
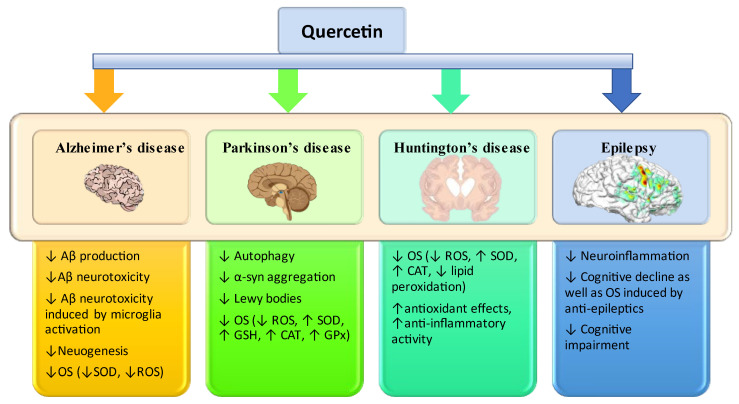
Diagram with possible neuroprotective effects of quercetin. Abbreviations: ↑, increase; ↓, decrease; Aβ, amyloid beta-peptide; CAT, catalase; GPx, superoxide dismutase; GSH, glutathione; OS, oxidative stress; ROS, reactive oxygen species; SOD, superoxide dismutase [14,16,47].

**Table 1 antioxidants-12-01418-t001:** Que content (mg 100 g^−1^ or mg 100 mL^−1^) in selected foods and beverages [18].

	Source			Que	References
Food	Common Name	Scientific Name	Active Portions	mg 100 g^−1^ Weight	
Fruits	Acerola	*Malpighia emarginata*	Fruits	4.74	[20]
	Apple	*Malus domestica*	Fruits	19.36	[21]
	Cranberry	*Vaccinium oxycoccus*	Fruits	25.0	[22]
	Apricots	*Prunus armeniaca*	Fruits	1.63	[23]
	Blackberries	*Rubus* spp.	Fruits	3.58	[24]
	Blueberries	*Vaccinium* spp.	Fruits	7.67	[24]
	Cherries	*Prunus avium*	Fruits	17.44	[25]
	Cranberries	*Vaccinium macrocarpon*	Fruits	14.84	[24]
	Grapefruit	*Citrus paradisi*	Fruits	0.50	[26]
	Grapes	*Vitis vinifera*	Fruits	3.7	[26]
Vegetables	Capers, raw	*Capparis spinosa*	Flower buds	233.84	[27]
	Onions, raw	*Allium cepa*	Bulbs	34.8	[26]
	Dill weed, fresh	*Anethum graveolens*	Leaves	74.5	[28]
	Oregano	*Origanum vulgare*	Leaves	42.00	[29]
	Tarragon, fresh	*Artemisia dracunculus*	Leaves	10.00	[26]
	Chicory	*Cichorium intybus*	Leaves	25.2	[30]
**Beverages**				**mg 100 mL^−1^**	
	Black tea			2.50	[31]
	Red wine			3.16	[32]

Source: Phenol Explorer and USDA Database for the Flavonoid Content of Selected Foods.

**Table 4 antioxidants-12-01418-t004:** Protective effects against oxidative stress and neuroinflammation induced by Que in vitro in the case of Parkinson’s disease.

Types of Que	Concentration	Model	Exposure	Effects	Ref.
Que	Dosage: 0.1 μM	Microglial (N9)-neuronal (PC12) cells	MPP	↓ iNOS gene expression, ↓ ROS, ↓ cellular death, ↓ DNA fragmentation, ↓apoptosis, ↓ nuclear translocation of apoptosis-inducing factor, ↓ caspase-3 activation	[102]
	Dosage: 10 mM	PC12 cells	α-Synuclein	↓ Aβ fibrillation	[105]
Isoquercetin	Dosage: 10, 50, and 100 μM	PC12 cells	6-OHDA	↓ ROS, ↑ SOD, ↑ GSH, ↑ CAT, ↑ GPx	[106]
Quercetin glycoside	Dosage: 10, 50, and 100 μM	PC12 cells	6-OHDA	↑ antioxidant activity, ↑ GSH, ↑ GPx	[107]

Abbreviations: ↑, increase; ↓, decrease; 6-OHDA, 6-hydroxydopamine; Aβ, amyloid beta-peptide; CAT, catalase; DNA, 6-hydroxydopamine; GPx, glutathione peroxidase; GSH, glutathione; iNOS, inducible nitric oxide synthase; MPP, 1-methyl-4-phenylpyridinium; LDH, lactate dehydrogenase; MMP, matrix metalloproteinases; OS, oxidative stress; Q3G, quercetin-3-glucuronide; Que, quercetin; ROS, reactive oxygen species; SOD, superoxide dismutase.

**Table 5 antioxidants-12-01418-t005:** Protective effects against oxidative stress and neuroinflammation of Que in vivo in the case of Parkinson’s disease.

Types of Que	Concentration	Model	Exposure	Effects	Ref.
Que	Dosage: 25 mg kg^−1^ Ad: p.o.	Wistar rats	Haloperidol MPTP	↓ cataleptic score, ↑ actophotometer activity score, ↑ GSH, ↓ lipid peroxidation, ↓ ROS	[108]
	Dosage: 25 and 50 mg kg^−1^ Ad: intragastrically Duration: 14 days	Wistar rats	MPTP	↓ TNF-α, ↓ IL-1β and ↓ IL-6, ↓ glutamate level,	[109]
	Dosage: 50, 100, and 200 mg kg^−1^ Ad: p.o. Duration: 14 days	adult male C57BL/6 mice	MPTP	↓ striatal dopamine depletion, ↓ level of acetylcholine, ↑ AchE activity, ↑ motor deficits, ↑ GPx, ↑ SOD	[110]
	Dosage: 100, 200, and 300 mg kg ^−1^ Duration: 14 days	Wistar rats	6-OHDA	↑ spatial memory, ↓ OS, ↓ AchE activity, ↑ antioxidant activity, ↓ neuronal damage	[111]
	Dosage: 20 mg kg^−1^ Ad: i.p. Duration: 1 month	Wistar rats	6-OHDA	↓ neuroplastic changes in neural circuits, ↓ excitability in neurons involved in epilepsy, ↓ NMDA receptor functionality	[112]
	Dosage: 25–75 mg kg^−1^ Duration: 12 h intervals for 4 days	Wistar rats	Rotenone	↓ nigral GSH depletion, ↓ ROS, ↓ striatal DA loss, ↑ mitochondrial complex, ↓ neuronal death	[113]
	Dosage: 50 mg kg^−1^ Ad: p.o. Duration: 14 days	Wistar rats	Rotenone	↑ AchE activity, ↑ SOD, ↓ GPx, ↓ CAT	[114]
Que + fish oil	Dosage: 25 mg kg^−1^ Ad: p.o. Duration: 28 days	Wistar rats	Rotenone	↑ mitochondrial functions, ↑ GSH, ↑ antioxidant defenses	[115]

Abbreviations: ↑, increase; ↓, decrease; 6-OHDA, 6-hydroxydopamine; AchE, acetylcholinesterase; CAT, catalase; DNA, deoxyribonucleic acid; GPx, glutathione peroxidase; GSH, glutathione; iNOS, inducible nitric oxide synthase; i.p., intraperitoneal; MPP, 1-methyl-4-phenylpyridinium; MMP, matrix metalloproteinases; NMDA, N-methyl-D-aspartate; OS, oxidative stress; p.o., oral; Q3G, quercetin-3-glucuronide; Que, quercetin; ROS, reactive oxygen species; SOD, superoxide dismutase.

**Table 6 antioxidants-12-01418-t006:** Protective effects against oxidative stress and neuroinflammation induced by Que in vivo in the case of Huntington’s disease.

Types of Que	Concentration	Model	Exposure	Effects	Ref.
Que	Dosage: 25 mg kg^−1^ Ad: p.o. Duration: 21 days	Wistar rats	3-NPA	↑ ATP, ↑ activity of complex II and V enzyme of respiratory chain complex, ↓ ROS, ↑ SOD, ↑ CAT, ↓ lipid peroxidation,	[117]
	Dosage: 25–50 mg kg^−1^ Ap: i.p. Duration: 4 days	Sprague Dawley rats	3-NPA	↓ gait despair, ↓ microglial proliferation, ↓ anxiety, ↑ astrocyte numbers in the lesion core, ↓ motor coordination deficits, ↓ serotonin metabolism	[118]
Que + lycopene	Dosage: 50 mg kg^−1^ Duration: 14 days	Wistar rats	3-NPA	↓ anxiety, ↓ depression	[119]
Que + fish oil	Dosage: 25 mg kg^−1^	Wistar rats	3-NPA	↓ OS, ↑ motor function	[120]
Que + sesamol	Dosage: 25, 50, and 100 mg kg^−1^ Ad: i.p. Duration: 14 days before and 14 days after QA administration	Wistar rats	QA	↓ behavioral, biochemical, and neurochemical alterations in the rat brain, ↑ antioxidant effects, ↑ anti-inflammatory activity	[121]

Abbreviations: ↑, increase; ↓, decrease; 3-NPA, 3-nitropropionic acid; CAT, catalase; GSH, glutathione; i.p., intraperitoneal; MPP, 1-methyl-4-phenylpyridinium; OS, oxidative stress; p.o., oral; QA, quinolinic acid; Que, quercetin; ROS, reactive oxygen species; SOD, superoxide dismutase.

**Table 7 antioxidants-12-01418-t007:** Protective effects against oxidative stress and neuroinflammation induced by Que in the case of epilepsy.

Types of Que	Concentration	Model	Type of Test	Exposure	Effects	Ref.
Que	Dosage: 5, 10, 20, and 40 mg kg^−1^	Albino rats	in vivo	PTZ	↑ antiseizure effect, ↑ anticonvulsant effect	[123]
	Dosage: 25, 50, and 100 mg kg^−1^ Ad: i.p.	Wistar rats	in vivo	PTZ	↑ anticonvulsant effects, ↓ seizure severity, ↓ lipid peroxidation, ↑ antioxidant effect, ↑ memory retrieval in the passive avoidance task	[125]
	Dosage: 10, 20, and 40 mg kg^−1^ Ad.: p.o. Duration: 15 days	Swiss albino mice	in vivo	PTZ	↑ immobility time, ↓ seizure severity	[126]
Que/ *Anisomelesma labarica*	Dosage: 25 and 50 mg kg^−1^ Ad: i.p.	Wistar rats	in vivo	PTZ	↓ locomotor activity and motor activity performance	[127]
	Dosage: 6.25 and 12.5 mg kg^−1^ Ad: i.p. Duration: 1 week	Wistar rats	in vivo	PTZ	potentiating the GABAergic system, inhibition of the NMDA receptor and Na^+^ channels.	

Abbreviations: ↑, increase; ↓, decrease; GABA, glutamate and γ-amino butyric acid; i.p., intraperitoneal; NMDA, N-methyl-D-aspartate; p.o., oral; PTZ, pentylenetetrazol; Que, quercetin.

## Data Availability

The datasets used and analyzed during the current study are available from the corresponding author upon request.

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
