# Peer review of "Exploitation of Quercetin’s Antioxidative Properties in Potential Alternative Therapeutic Options for Neurodegenerative Diseases"

_antioxidants, 2023, doi:10.3390/antiox12071418_

Round 1

Reviewer 1 Report

Comments and suggestions

1. In the introduction table 1, insert one column and add references.

2. Line 91-92, Quercetin is a more potent antioxidant than other antioxidant nutrients, such as vitamin C, vitamin E, and β-carotene [20]. Underline is necessary, please check and confirm.

3.  Methodology, please use a Prisma flow chart for methodology.

4. Figure 2, QUERCETIN, all capital is necessary, please check and confirm.

5. In the case of abbreviation first time use the short form with the abbreviation then use only the short form throughout the manuscript.

6. Tables 4, 5 & 6, in the figure legend use all abbreviations that used short form in the table.

7. Please check Table 6. Protective effects against oxidative stress and neuroinflammation, induced by quercetin in vitro in the case of Huntington’s disease. It will be in vivo or in vitro?

8. Extend 3.4. Epilepsy this section with the relevant study.

9.  Please check carefully all grammatical and typo errors throughout the manuscript.

Minor editing of English language required

Author Response

Comments and suggestions

  1. In the introduction table 1, insert one column and add references.

Response: We added the references.

  1. Line 91-92, Quercetin is a more potent antioxidant than other antioxidant nutrients, such as vitamin C, vitamin E, and β-carotene[20]. Underline is necessary, please check and confirm.

Response: No, it is not necessary, we modified it.

  1. 3Methodology, please use a Prisma flow chart for methodology.

Response: We used the flow chart prism for the methodology. And the "Methodology" chapter is revised: “Search Strategy"

The current systematic review was conducted following the Preferred Reporting Items for Systematic Reviews and Meta-Analysis (PRISMA) guidelines [58], employing several electronic databases (Science Direct, PubMed, and Google Scholar) and using the following keywords: ((quercetin[Title/Abstract]) AND (Alzheimer's disease[Title/Abstract])) AND (amyloid[Title/Abstract]); (quercetin[Title/Abstract]) AND (Parkinson's disease[Title/Abstract]); (quercetin[Title/Abstract]) AND (Huntington's disease[Title/Abstract]); (quercetin[Title/Abstract]) AND (pentylenetetrazol [Title/Abstract]). Inclusion criteria were included experimental studies (in vivo and in vitro) until May 2023 in English, which evaluated the antioxidant properties of quercetin in potential alternative therapeutic options of neurodegenerative diseases.

Exclusion Criteria

We applied the following exclusion criteria: 1) conference abstracts, books, book chapters, and unpublished results; 2) non-English papers; 3) reviews, systematic re-views, meta-analysis, and letters.

Data Extraction

Among the initial 868 reports that were collected through electronic search, 554 were omitted due to duplicated results, 47 were ruled out because of the article type, 190 review articles were omitted, and were deemed irrelevant based on abstract and/or title information. Besides, 1 was excluded because they were not in English language.

Data Synthesis

Finally, 43 articles were included in this study, as demonstrated in a diagram of the literature search and selection process (see Figure 2). It was thought that the studies would be too heterogeneous to be combined. Therefore, a narrative synthesis was performed. The results are summarized according to the type of neurodegenerative disease (AD, PD, HD and epilepsy) and the test (in vivo, in vitro).

Figure 2. The PRISMA flow chart of the selection process for the included studies.”

  1. Figure 2, QUERCETIN, all capital is necessary, please check and confirm.

Response: No, it is not necessary, we modified it.

  1. In the case of abbreviation first time use the short form with the abbreviation then use only the short form throughout the manuscript.

Response: We checked and modified.

  1. Tables 4, 5 & 6, in the figure legend use all abbreviations that used short form in the table.

Response: We checked and modified.

  1. Please check Table 6.Protective effects against oxidative stress and neuroinflammation, induced by quercetin in vitro in the case of Huntington’s disease. It will be in vivo or in vitro?

Response: Corrected. The title of the table is “Protective effects against oxidative stress and neuroinflammation, induced by quercetin in vivo in the case of Huntington’s disease”.

  1. Extend 3.4. Epilepsy this section with the relevant study.

Response: We have extended this section. Revised section: “Epilepsy is a neurological disorder characterized by recurrent spontaneous seizures, being caused by an imbalance in excitatory and inhibitory neurotransmission [126]. Glutamate and γ-amino butyric acid (GABA) are the major excitatory and inhibitory neurotransmitters in the CNS [127]. A GABA receptor antagonist is pentylenetetrazol (PTZ), which is used to create a chemically induced seizure model in animals [127]. The frequency and severity of these recurrent seizures can vary. Thus, a low dose of Que (25 mg kg -1) can reduce the number and duration of spike-wave discharges in WAG/Rij rats [128]. In addition, a reduction in the levels of TNF-alpha, IL-6 and NO was observed compared to the control group.

In a study using PTZ-induced seizure model rats, Que administration at 10 mg kg-1 intraperitoneally 30 minutes before PTZ injection significantly prolonged the onset and reduced the severity of the seizure, but, at an increased concentration of 40 mg kg-1, Que failed to prevent the effects of PTZ [127]. Also, Nassiri-Asl et al., [129] showed that administration of 35 mg kg-1 PTZ after 50 mg kg-1 Que reduces seizure severity during kindling and improves performance in a passive avoidance task in kindled rats. Also, quercetin supplementation of levetiracetam improved depression that is associated with epilepsy, decreased immobility time and reduced seizure severity [130]. In addition, Choudhary et al., 2011, [131] isolated and evaluated the antiepileptic potential of the flavonoid fractions of Anisomeles malabarica leaves, both acute and chronic. In the acute treatment (25 and 50 mg kg-1, i.p.) with toxic effects observed, respectively chronic treatment for one week (6.25 and 12.5 mg kg-1, i.p.) a significant antiepileptic effect was observed without causing side effects neurotoxic [131].

  1. Please check carefully all grammatical and typo errors throughout the manuscript.

Response: Manuscript has been checked by a colleague fluent in English writing. The grammar, spelling, punctuation, and phrasing of the paper were improved in order to enhance its readability.

Reviewer 2 Report

The manuscript by Rarinca and co-authors is an overview of neuroprotective effects of quercetin in Alzheimer’s disease (AD), 29 Parkinson’s disease (PD), Huntington's disease (HD) and epilepsy. The article is well written, detailed and supported by many references.

However, the manuscript needs to be improved in several parts:

Can quercetin cross the blood brain barrier? It should be discuss.

Line 71: does “quercetin can inhibit the formation of amyloid-ß proteins in the fibrils” mean “quercetin can inhibit the formation of amyloid-ß fibrils”?

Line 96: reference 1 is not appropriate

Line 144: “Thus, figure 2, we highlighted the main enzymes that have the role…” There are only a few enzymes highlighted in the figure.

Line 180: figure 1 should be probably indicated instead of figure 2

Table 2:

Reference 58: Hek cells have been used

Reference 61: cells have been treated with quercetin and not Quercetin-3′-glucoside as indicated; CREB/BDNF signaling pathway has not been studied

Reference 64: should be fully rechecked (cells, treatment…)

Reference 65: “ad: i.p.” please correct

Table 3:

Reference 68 and 74: These are in vitro studies, they should be in table 2

Line 271: It is not clear if quercetin has any effect on type 2 diabetes. If it has no effect, why did you discuss the link between AD and type 2 diabetes?

Line 290: probably the sentence should be “mRNA and protein levels”

Line 293: probably the sentence should be “Thus, the formation of α-Syn fibrils” or “Thus, α-Syn fibrillation”

Line 298: on the phenyl ring at positions 2, 4 and 5 there are hydroxyl groups, not hydrogens

Table 4:

Reference 92: there is a decrease of apoptosis, please correct the arrow

Reference 95: is a cell free system

Line 327: reference 106 is not appropriate

Line 361: ‘in vitro’ should be replaced with ‘in vivo’

Line 371: reference 111 should be replaced with reference 115

Only minor editing is required

Author Response

The manuscript by Rarinca and co-authors is an overview of neuroprotective effects of quercetin in Alzheimer’s disease (AD), 29 Parkinson’s disease (PD), Huntington's disease (HD) and epilepsy. The article is well written, detailed and supported by many references.

However, the manuscript needs to be improved in several parts:

Can quercetin cross the blood brain barrier? It should be discuss.

Response: Yes, we mentioned that in the phrase: “Additionally, Q3G has been found to specifically localize in human brain tissue, suggesting that it may be able to cross the blood-brain barrier and reach areas of the brain affected by AD [64].” Also, we added a phrase about the mechanism. “The mechanism by which Que can cross the BBB is through passive diffusion, this is a process that allows small molecules to move across the BBB from an area of high concentration to an area of low concentration [66].”

Line 71: does “quercetin can inhibit the formation of amyloid-ß proteins in the fibrils” mean “quercetin can inhibit the formation of amyloid-ß fibrils”?

Response: Yes. The formation of amyloid-ß fibrils is a result of the aggregation of amyloid-ß proteins, and quercetin has been found to inhibit this process by binding to the proteins and modifying their aggregation behavior.

Line 96: reference 1 is not appropriate

Response: We have replaced it with an appropriate reference.

Line 144: “Thus, figure 2, we highlighted the main enzymes that have the role…” There are only a few enzymes highlighted in the figure.

Response: Corrected. Revised sentence: Thus, figure 2, we highlighted some enzymes that have the role of protecting the body from oxidative stress and have neuroprotective effects in the case of neurodegenerative diseases.

Line 180: figure 1 should be probably indicated instead of figure 2

Response: Replaced “figure 2” with “figure 1”.

Table 2:

Reference 58: Hek cells have been used.

Response: Replaced with “human embryonic kidney cells”.

Reference 61: cells have been treated with quercetin and not Quercetin-3′-glucoside as indicated; CREB/BDNF signaling pathway has not been studied

Response: We rechecked the reference. Revised line:

Que

Dosage: 10, 20, 40, and 80 μmol L-1

Duration: 24 h, 48 h, and 72 h

PC12 cells      

25–35

↑ the survival rate of PC12 injured by Aβ25-35, promote cell proliferation, and antagonize the toxicity of Aβ, ↓ROS

Reference 64: should be fully rechecked (cells, treatment…)

Response: We rechecked the reference. Revised line:

Q3G

Dosage: 25 μmol L-1

Tg2576 AD primary neuron cultures

1–40, 1–42

↑neuronal survival, ↑c-Jun N-terminal kinases, ↓stress-induced impairments

Reference 65: “ad: i.p.” please correct

Response: We rechecked the reference, and the method of administration is not mentioned in the article, and we deleted it: “ad: i.p.”.

Table 3:

Reference 68 and 74: These are in vitro studies, they should be in table 2

Response: We moved them to the table 2.

Line 271: It is not clear if quercetin has any effect on type 2 diabetes. If it has no effect, why did you discuss the link between AD and type 2 diabetes?

Response: We deleted this part, as the link between type 2 diabetes and Alzheimer's disease is not yet well understood.

Line 290: probably the sentence should be “mRNA and protein levels”

Response: Corrected. Revised sentence: On the other hand, Que administration protects microglia cells against MPP+- induced increases in mRNA and protein levels of IL-1, IL-6 and TNF-α, due to its antioxidant action.

Line 293: probably the sentence should be “Thus, the formation of α-Syn fibrils” or “Thus, α-Syn fibrillation”

Response: Replaced with “Thus, the formation of α-Syn fibrils”. Revised sentence: Thus, the formation of α-Syn fibrils can be inhibited by Que and oxidized Que through their 1:1 covalent binding.

Line 298: on the phenyl ring at positions 2, 4 and 5 there are hydroxyl groups, not hydrogens.

Response: Corrected. Revised sentence: It is a hydroxylated analogue of dopamine and is a benzenetriol with hydroxyl groups on the phenyl ring at positions 2, 4 and 5.

Table 4:

Reference 92: there is a decrease of apoptosis, please correct the arrow

Response: Corrected.

Reference 95: is a cell free system

Response: Corrected.

Line 327: reference 106 is not appropriate

Response: We have replaced it with an appropriate reference.

Line 361: ‘in vitro’ should be replaced with ‘in vivo’

Response: Replaced ‘in vitro’ with ‘in vivo’.

Line 371: reference 111 should be replaced with reference 115

Response: Replaced reference 111 with reference 115.

Round 2

Reviewer 1 Report

Thanks for revised your manuscript. Good luck.